# Expressional artifact caused by a co-injection marker *rol-6* in *C. elegans*

HoYong Jin[1], Scott W. Emmons[2]*, Byunghyuk Kim[1]*

**1** Department of Life Science, Dongguk University-Seoul, Goyang, Republic of Korea, **2** Department of Genetics and Dominick P. Purpura Department of Neuroscience, Albert Einstein College of Medicine, Bronx, New York, United States of America

* bkim12@dongguk.edu (BK); scott.emmons@einstein.yu.edu (SE)

**Data Availability Statement:** All relevant data are within the manuscript and its Supporting Information files.

**Funding:** Some strains were provided by the CGC, which is funded by NIH Office of Research Infrastructure Programs (P40 OD010440). This

## Abstract

In transgenic research, selection markers have greatly facilitated the generation of transgenic animals. A prerequisite for a suitable selection marker to be used along with a test gene of interest is that the marker should not affect the phenotype of interest in transformed animals. One of the most common selection markers used in *C. elegans* transgenic approaches is the *rol-6* co-injection marker, which induces a behavioral roller phenotype due to a cuticle defect but is not known to have other side effects. However, we found that the *rol-6* co-injection marker can cause expression of GFP in the test sequence in a male-specific interneuron called CP09. We found that the *rol-6* gene sequence included in the marker plasmid is responsible for this unwanted expression. Accordingly, the use of the *rol-6* co-injection marker is not recommended when researchers intend to examine precise expression or perform functional studies especially targeting male *C. elegans* neurons. The *rol-6* sequence region we identified can be used to drive a specific expression in CP09 neuron for future research.

## Introduction

Efficient transgenic techniques are used in various model systems to detect gene expression and assess genetic function. In the nematode *Caenorhabditis elegans*, for example, gene expression can be monitored using transgenic worms generated by a simple, gonadal microinjection of a plasmid that drives GFP expression under the control of a promoter for a gene of interest [1]. During the course of the DNA transformation procedure, one easy way to select transformed animals is by using easily-detected co-injection markers. In *C. elegans*, several co-injection markers are commonly used, which include visible fluorescent markers (e.g. *ttx-3p*::*GFP*, *myo-2p*::*mCherry*) [2, 3] and rescuing markers that restore lethal or non-lethal phenotypes (e.g. *pha-1*, *unc-119*, *dpy-5*) [4–6]. One type of dominant selectable marker, *rol-6(su1006)*, is widely used, because it shows a dominant roller phenotype that is easily observed and can be used in a wild type background [7, 8].

A prerequisite for the use of co-injection markers is that the phenotype induced by the co-injection marker must not interfere with expression or scoring of the gene being tested. In this

work was supported by the National Research Foundation of Korea (NRF) grant funded by the Korea government (MSIT) (2018R1C1B5043569) (to BK) and by the United States National Institutes of Health (R01MH112689) (to SE). The funders had no role in study design, data collection and analysis, decision to publish, or preparation of the manuscript.

**Competing interests:** The authors have declared that no competing interests exist.

study, we report that the widely-used *rol-6* marker unexpectedly activates the test gene in a male interneuron called CP09 in *C. elegans*. This unwanted expression could potentially result in misidentification of cell types in a gene expression study as well as affect the results of functional studies that utilize *rol-6* as a co-injection marker.

## Results and discussion

During the course of experiments to determine the tissue-specific expression pattern of 10 putative synaptic genes in *C. elegans*, we generated transgenic worms with promoter::GFP test genes using *rol-6(su1006)* as a co-injection marker (for information on the genes and the promoter fragments assayed, see S1 Table). We noticed that in eight out of 10 transgenic lines, GFP was expressed in the CP09 neuron among other diverse neurons [9]. CP09 is a male-specific interneuron located in the pre-anal ganglion of the male tail that forms chemical and electrical synapses with many other male-specific or sex-shared neurons (Fig 1). Male *C. elegans* have 10 CP ventral cord neurons (CP00~CP09) [10]. The CP neurons are believed to have similar properties due to a similar developmental origin, in which all CPs are generated from Pn. aapp cells, but some CP neurons are reported to use different neurotransmitters [10–13]. Interestingly, out of the eight transgenic lines showing CP09 expression, four of the transgenes were expressed in most or many neurons (i.e. many CP neurons), but the other four were expressed exclusively in CP09 among the 10 CP neurons. Therefore, we suspected that the CP09 expression may be an expression artifact.

It is widely known that GFP reporters driven by diverse promoters often show artificial fluorescence in posterior gut cells, in several muscle cells, and even in one neuron called PVT [15, 16]. One potential cause of these artifacts was suggested to be an effect of the *unc-54* 3′ UTR, which is attached to the GFP coding sequence in most *C. elegans* vectors [16]. To test whether the *unc-54* 3′ UTR can also cause expression in CP09, we replaced the *unc-54* 3′ UTR

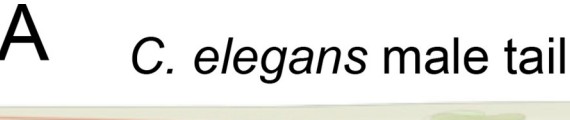

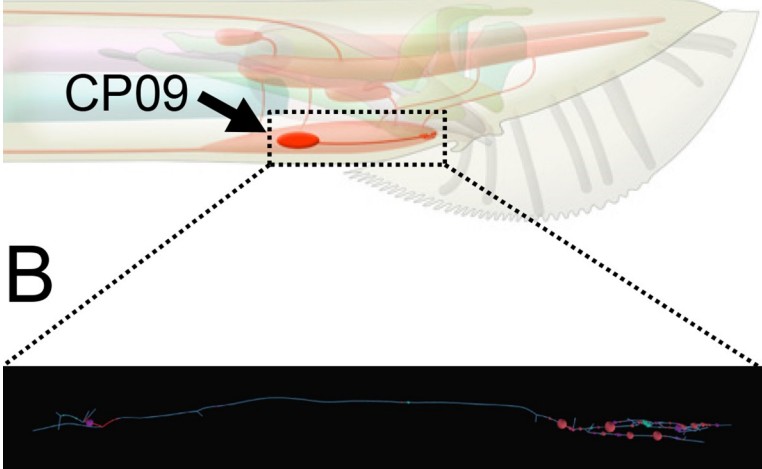

**Fig 1. The CP09 neuron in *C. elegans*.** (A) Schematic of the position of cell body and axon of CP09 in the pre-anal ganglion of the *C. elegans* male tail. Reprinted from [14] under a CC BY license, with permission from WormAtlas (https://www.wormatlas.org/), original copyright 2019. (B) A skeleton map of CP09. Dots indicate presynapses (pink), postsynapses (purple), and gap junctions (blue). Information of individual synapses is accessible at WormWiring (http://wormwiring.org/).

with the *let-858* 3′ UTR in a promoter::GFP fusion for one of the test genes that showed exclusive expression in CP09 among the 10 CP neurons (*T19A6.4* gene). When transgenic animals were generated by microinjection of the *T19A6.4p*::*GFP*::*let-858 3' UTR* fusion along with the *rol-6* co-injection marker, they still showed CP09 expression, suggesting that at least for this gene factors other than the *unc-54* 3′ UTR are likely involved in the generation of the CP09 signal.

The second possibility was that the *rol-6* co-injection marker used in the microinjection procedure caused the expression in CP09. To test this idea, we injected an empty GFP vector (pPD95.75), which contains no promoter for the GFP coding sequence, together with the *rol-6* co-injection marker (pRF4). The resulting transgenic animal showed a robust GFP expression in CP09 (Fig 2A). However, when the empty GFP vector was injected with another co-injection marker *ttx-3p*::*GFP* (expressed in AIY neuron in the head), the CP09 signal was not observed (Fig 2B). Thus, the *rol-6* co-injection marker itself can promote transcription in the CP09 neuron.

Homologous recombination between co-injected DNA molecules contributes to the formation of stable extrachromosomal arrays [8]. Most *C. elegans* vectors have a backbone based on the pUC19 plasmid, and thus have high sequence similarities that are potentially utilized for homologous recombination during extrachromosomal arrays formation. For example, the pRF4 and pPD95.75 plasmids share ~2.5 kb sequences that include the *E.coli* ampicilin

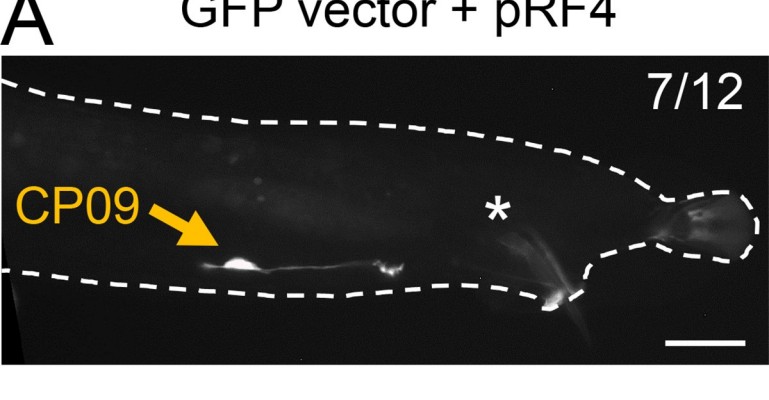

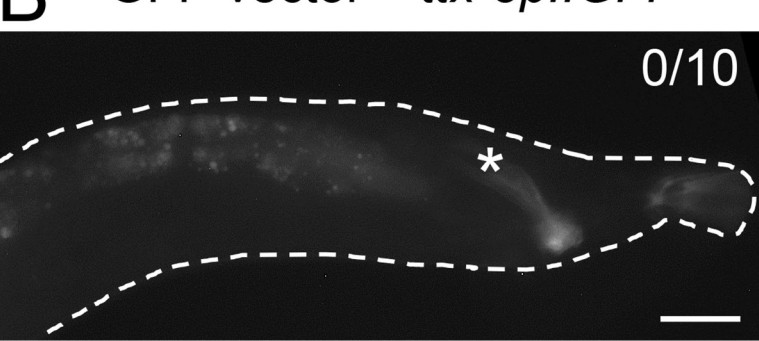

**Fig 2. The *rol-6* co-injection marker causes an expressional artifact in CP09.** GFP expression in the male tail of transgenic worms injected with an empty GFP vector (pPD95.75) along with *rol-6(su1006)* plasmid (pRF4) (A) or *ttx-3p*::*GFP* plasmid (B). Exclusive CP09 expression was observed in seven out of 12 independent transgenic lines injected with GFP vector + pRF4 (7/12), whereas no CP09 expression was observed in 10 independent lines with GFP vector + *ttx-3p*::*GFP* (0/10). Asterisks indicate autofluorescence in the spicule. Scale bar, 20 μm.

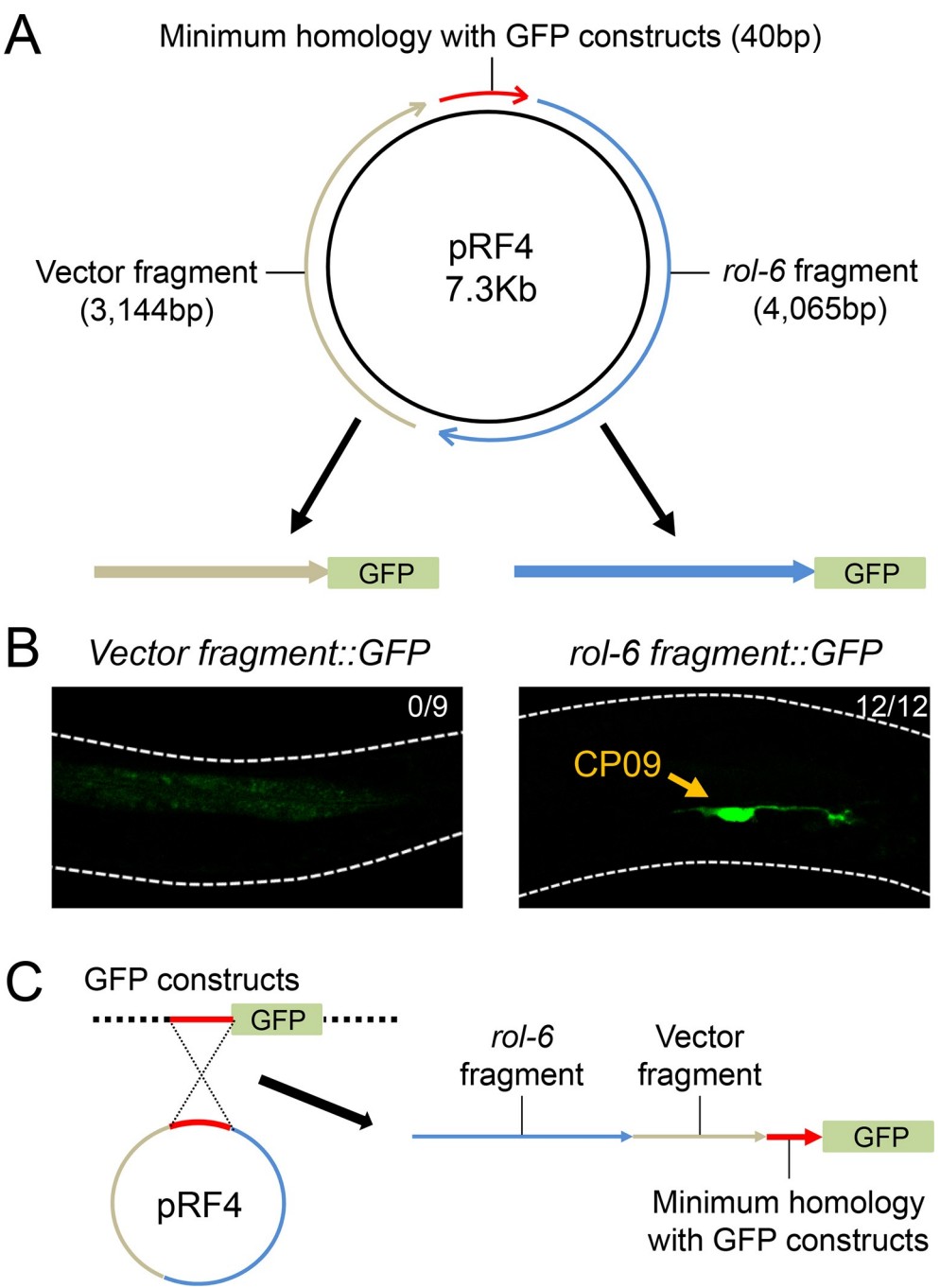

**Fig 3. The *rol-6* fragment of pRF4 is responsible for CP09 expression.** (A) Schematic of cloning procedure to identify a region of pRF4 plasmid responsible for CP09 expression. Either *rol-6* or vector fragment was subcloned into the empty GFP vector pPD95.75 and the resulting plasmids were injected to generate transgenic worms. (B) CP09 expression was observed in all 12 independent transgenic lines injected with *rol-6 fragment::GFP* (12/12), whereas no CP09 expression was observed in nine independent lines with *vector fragment::GFP* (0/9). (C) Proposed model of homologous recombination between pRF4 plasmid and GFP constructs.

resistance gene and origin of replication. We also found a 40 bp homology shared between the pRF4 plasmid and GFP constructs generated by promoter::GFP fusion PCR, which constitutes the multiple cloning site of the vectors (minimum homology with GFP constructs) (Fig 3A).

The minimum homology sequence is located upstream of the GFP coding region of pPD95.75 and included in most promoter::GFP fusion constructs, as this inclusion ensures that an artificial intron is placed in front of the GFP sequence for efficient reporter gene expression [16]. The full sequence information of pRF4 is available in S1 Text. (Although pRF4 has been used widely in the *C. elegans* research community, the accurate pRF4 sequence, to our knowledge, is not yet available in public.)

To find a region responsible for CP09 expression, we divided the pRF4 plasmid except for the minimum homology region into two fragments, namely "*rol-6*" and "vector" fragments, and cloned these into the empty GFP vector pPD95.75 (Fig 3A). When the *rol-6 fragment*::*GFP* was injected, the resulting transgenic animal showed a robust GFP expression in CP09 (Fig 3B). However, we could not observe any GFP expression in CP09 when the *vector fragment*::*GFP* was used for injection (Fig 3B). Therefore, it is likely that when a GFP construct is injected together with the *rol-6* co-injection marker pRF4, the *rol-6* fragment fused to GFP by homologous recombination generates unwanted transcription and GFP expression in CP09 (Fig 3C).

Our results raise an obvious problem in using the *rol-6* co-injection marker for gene expression or functional studies especially on the *C. elegans* male, as this marker can induce unwanted expression in the male CP09 neuron. The *rol-6* fragment of pRF4 likely contains a driver sequence that triggers CP09 expression. For expression studies using the *rol-6* co-injection marker, any CP09 expression needs to be double-checked by using another type of co-injection marker. For functional studies, it should be determined whether the use of the *rol-6* marker affected interpretation of the results. For example, techniques called GRASP (GFP reconstitution across synaptic partners) and iBLINC (in vivo biotin labelling of intercellular contacts) have been developed to visualize synapses formed between specific pairs of neurons that are defined by cell-specific drivers [17, 18]. Since CP09 has many synaptic connections with other male-specific and sex-shared neurons (see Fig 1), artifactual expression in CP09 can potentially generate additional synapse signal when using GRASP or iBLINC. We recommend not using the *rol-6* marker if studies are designed to examine gene expression or function in the male tail of *C. elegans*.

To avoid the unwanted CP09 expression, homologous recombination between the *rol-6* co-injection marker and any expression constructs may be minimized by reducing their sequence homologies. For example, fusion PCR-based promoter::GFP constructs share a minimum 40 bp homology with the pRF4 plasmid. If a promoter::GFP construct is designed to omit the homologous sequences, it may be possible to suppress CP09 expression caused by the *rol-6* co-injection marker. However, it will be difficult to test this idea using plasmid-based GFP constructs, as most *C. elegans* plasmids share a backbone and usually have a high sequence homology [16].

Cell-specific promotors or drivers are invaluable tools for transgenic research, because they allow us to confine gene expression to subsets of cells or even to a specific cell. Several such drivers have been identified and used extensively in the *C. elegans* community [19]. In this study, we identified that the *rol-6* fragment of pRF4 drives expression in the male-specific CP09 neuron. This sequence can be used as a CP09-specific driver for future research.

## Materials and methods

### *C. elegans* maintenance

CB4088 *him-5(e1490)* worms were used as the wild-type reference strain to generate worm populations containing large numbers of males. Worms were grown at 20°C on standard nematode growth media (NGM) plates with OP50 *E. coli* as a food source and maintained according to standard methods [20].

## Transgenic strains and molecular cloning

To obtain transgenic worms, plasmids or fusion PCR products [21] were injected into *him-5 (e1490)* worms at ~50 ng/ μl with co-injection marker pRF4 (*rol-6(su1006)*) or *ttx-3p*::*GFP* at 50 ng/ μl.

*T19A6.4p*::*GFP*::*let-858 3′ UTR* fusion was obtained by a PCR-fusion method [21]. *T19A6.4p* was PCR-amplified from N2 worms as described previously [9] and then fused to *GFP*::*let-858 3′ UTR* amplified from pPD135.02 vector (a gift from Andrew Fire) to generate *T19A6.4p*::*GFP*::*let-858 3′ UTR* PCR fragment.

To generate *rol-6 fragment*::*GFP*, the *rol-6* fragment (4,065 bp) was PCR-amplified from pRF4 with restriction sites of *SphI* and *XmaI* (primer F: 5´- AAAGGCATGC ttatcatcttcggtttt-gataaa-3´ and primer R: 5´- AACCCCGGG gtattcaaagcaggagaagc-3´). This PCR product was digested and ligated into *SphI*/*XmaI*-digested pPD95.75 vector.

To generate *vector fragment*::*GFP*, the vector fragment (3,144 bp) was PCR-amplified from pRF4 with restriction sites of *SphI* and *XmaI* (primer F: 5´- GGGGGCATGC gccctatagt gagtcgtatt-3´ and primer R: 5´- AACCCCGGG tttgttccctttagtgaggg-3´). This PCR product was digested and ligated into *SphI*/*XmaI*-digested pPD95.75 vector.

## Microscopy

Worms were prepared and imaged as described previously [22]. Briefly, 1-day-old males were mounted on 5% agar pads on glass slides using 10~50 mM sodium azide. Worms were observed with fluorescence microscopy (Zeiss Axio Imager.Z2) or confocal microscopy (Nikon Eclipse Ti). Images were processed using AxioVision (Zeiss) or NIS-Elements (Nikon). Figures were prepared using ImageJ software.

## Supporting information

**S1 Table. Promoter::GFP test genes and the fragments assayed for CP09 expression.**
(DOCX)

**S1 Text. pRF4 sequence (7,271 bp).**
(DOCX)

## Acknowledgments

We would like to thank Dr. David Hall for permission to use the CP09 image from WormAtlas. Some strains were provided by the CGC, which is funded by NIH Office of Research Infrastructure Programs (P40 OD010440). This work was supported by the National Research Foundation of Korea (NRF) grant funded by the Korea government (MSIT) (2018R1C1B5043569) (to BK) and by the United States National Institutes of Health (R01MH112689) (to SE).

## Author Contributions

**Conceptualization:** Scott W. Emmons, Byunghyuk Kim.

**Formal analysis:** HoYong Jin, Byunghyuk Kim.

**Funding acquisition:** Scott W. Emmons, Byunghyuk Kim.

**Supervision:** Scott W. Emmons, Byunghyuk Kim.

**Validation:** HoYong Jin.

**Visualization:** HoYong Jin, Byunghyuk Kim.

**Writing – original draft:** Byunghyuk Kim.

**Writing – review & editing:** Scott W. Emmons, Byunghyuk Kim.

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
