## [Decision Letter · Decision Letter 0]

6 Nov 2019

PONE-D-19-28634

Expressional artifact caused by a co-injection marker rol-6 in C. elegans

PLOS ONE

Dear Dr. Kim,

Thank you for submitting your manuscript to PLOS ONE. After careful consideration, we feel that it has merit but does not fully meet PLOS ONE’s publication criteria as it currently stands. Therefore, we invite you to submit a revised version of the manuscript that addresses the points raised during the review process.

We would appreciate receiving your revised manuscript by Dec 21 2019 11:59PM. To enhance the reproducibility of your results, we recommend that if applicable you deposit your laboratory protocols in protocols.io, where a protocol can be assigned its own identifier (DOI) such that it can be cited independently in the future. For instructions see: http://journals.plos.org/plosone/s/submission-guidelines#loc-laboratory-protocols

We look forward to receiving your revised manuscript.

Kind regards,

Denis Dupuy, Ph.D.

Academic Editor

PLOS ONE

Journal Requirements:

2.  We note that Figure(s) [1A] in your submission contain copyrighted images. All PLOS content is published under the Creative Commons Attribution License (CC BY 4.0), which means that the manuscript, images, and Supporting Information files will be freely available online, and any third party is permitted to access, download, copy, distribute, and use these materials in any way, even commercially, with proper attribution. For more information, see our copyright guidelines: http://journals.plos.org/plosone/s/licenses-and-copyright.

1.         You may seek permission from the original copyright holder of Figure(s) [1A] to publish the content specifically under the CC BY 4.0 license.

Reviewers' comments:

Reviewer's Responses to Questions

**Comments to the Author**

1. Is the manuscript technically sound, and do the data support the conclusions?

Reviewer #1: Yes

2. Has the statistical analysis been performed appropriately and rigorously? 

Reviewer #1: N/A

3. Have the authors made all data underlying the findings in their manuscript fully available?

Reviewer #1: Yes

4. Is the manuscript presented in an intelligible fashion and written in standard English?

Reviewer #1: Yes

5. Review Comments to the Author

Reviewer #1: This manuscript describes a relatively minor finding, but that finding is described in a concise manner. Furthermore, the implications for others working on C. elegans male tail biology are significant and this justifies publication as the information should be in the public domain. It is important to be aware of the presence of a cell specific enhancer in a common transformation marker as this could affect conclusions drawn from experiments.

The manuscript is very well put together. A few minor issues are identified below.

Line 51. Information on the original genes and the fragments assayed ought to be included perhaps in a supplementary table.

Line 94 – 101. Some clarification of the cloning used to identify what part of pRF4 is responsible for driving expression in CP09 is needed in the Results and Discussion text, in addition to that provided in the Materials and Methods section. It is not clear from the general description of sub-cloning of pRF4, if the minimum gfp homology region is in the vector or insert fragment. Also, cloning of the pRF4 vector fragment into pPD95.77, would lead to a repeat of the pUC vector parts in the same plasmid which would be expected to be unstable in E. coli and I would be surprised if this really was achieved.

Methods. Restriction enzyme names should include italics.

References. Gene and species names should be italicised.

Figs. 1 and 2. Reference is to “CP09” throughout the paper apart from in Figs 1 and 2 where it is “CP9”.

Fig. 3. Title for panel C is in square brackets and is not needed.

6. PLOS authors have the option to publish the peer review history of their article (what does this mean?). If published, this will include your full peer review and any attached files.

Reviewer #1: No

---

## [Author Response · Author response to Decision Letter 0]

14 Nov 2019

Jin et al.

RESPONSE TO REVIEWERS

Journal Requirements:

We have revised the text to meet the style requirements.

2. We note that Figure(s) [1A] in your submission contain copyrighted images. All PLOS content is published under the Creative Commons Attribution License (CC BY 4.0), which means that the manuscript, images, and Supporting Information files will be freely available online, and any third party is permitted to access, download, copy, distribute, and use these materials in any way, even commercially, with proper attribution. For more information, see our copyright guidelines: http://journals.plos.org/plosone/s/licenses-and-copyright.

1. You may seek permission from the original copyright holder of Figure(s) [1A] to publish the content specifically under the CC BY 4.0 license.

We have uploaded the completed Content Permission Form. Also, we have added the text in the figure caption for copyright information.

As the result was a none-or-all event and negative in a test, we think it can be shown as a simple description without mentioning about data availability. Thus, we have removed the phrase “data not shown”.

Review Comments to the Author

Reviewer #1: This manuscript describes a relatively minor finding, but that finding is described in a concise manner. Furthermore, the implications for others working on C. elegans male tail biology are significant and this justifies publication as the information should be in the public domain. It is important to be aware of the presence of a cell specific enhancer in a common transformation marker as this could affect conclusions drawn from experiments.

The manuscript is very well put together. A few minor issues are identified below.

Line 51. Information on the original genes and the fragments assayed ought to be included perhaps in a supplementary table.

The original information was described in Experimental Procedures section in our previous paper. To avoid repetition and clearly inform the source of information, we have indicated the source in the text:

“for information on the genes and the promoter fragments assayed, see Experimental Procedures section in [ref]”.

Line 94 – 101. Some clarification of the cloning used to identify what part of pRF4 is responsible for driving expression in CP09 is needed in the Results and Discussion text, in addition to that provided in the Materials and Methods section. It is not clear from the general description of sub-cloning of pRF4, if the minimum gfp homology region is in the vector or insert fragment. 

We agree that our description about the minimum gfp homology region was not clear. The minimum homology region is located upstream of the GFP coding region of pPD95.75. In the sub-cloning procedure, the minimum homology region was excluded so that both the vector and rol-6 fragments did not contain the minimum homology sequence. 

To clarify this, we added the following text in Results and Discussion section:

“The minimum homology sequence is located upstream of the GFP coding region of pPD95.75 and included in most promoter::GFP fusion constructs, as this inclusion ensures that an artificial intron is placed in front of the GFP sequence for efficient reporter gene expression [ref].”

“…we divided the pRF4 plasmid except for the minimum homology region into two fragments,…”

Also, cloning of the pRF4 vector fragment into pPD95.77, would lead to a repeat of the pUC vector parts in the same plasmid which would be expected to be unstable in E. coli and I would be surprised if this really was achieved.

We re-checked the sub-cloning of the vector fragment::GFP and found no errors in making it. This construct may be unstable due to the repetition of vector sequences, but we did not find any difficulties to sub-clone and amplify this construct using E. coli.

Methods. Restriction enzyme names should include italics.

We have revised the text accordingly.

References. Gene and species names should be italicised.

We have revised the text accordingly.

Figs. 1 and 2. Reference is to “CP09” throughout the paper apart from in Figs 1 and 2 where it is “CP9”.

We have revised the figures accordingly.

Fig. 3. Title for panel C is in square brackets and is not needed.

We have removed the title for panel C in Figure 3.

---

## [Editor Report · Decision Letter 1]

15 Nov 2019

PONE-D-19-28634R1

Expressional artifact caused by a co-injection marker rol-6 in C. elegans

PLOS ONE

Dear Dr. Kim,

Thank you for submitting your revised manuscript to PLOS ONE. This revised version of the manuscript addresses most the points raised during the initial review process. However, it would be preferable if you included the methodological information referenced within this article rather than as a call back :

“for information on the genes and the promoter fragments assayed, see Experimental Procedures section in [ref]To avoid repetition and clearly inform the source of information, we have indicated the source in the text: 

“for information on the genes and the promoter fragments assayed, see Experimental Procedures section in [ref]

This would allow readers to access all the pertinent information without juggling between two references.

We would appreciate receiving your revised manuscript by Dec 30 2019 11:59PM. To enhance the reproducibility of your results, we recommend that if applicable you deposit your laboratory protocols in protocols.io, where a protocol can be assigned its own identifier (DOI) such that it can be cited independently in the future. For instructions see: http://journals.plos.org/plosone/s/submission-guidelines#loc-laboratory-protocols

We look forward to receiving your revised manuscript.

Kind regards,

Denis Dupuy, Ph.D.

Academic Editor

PLOS ONE

---

## [Author Response · Author response to Decision Letter 1]

17 Nov 2019

Jin et al.

RESPONSE TO REVIEWERS

Thank you for submitting your revised manuscript to PLOS ONE. This revised version of the manuscript addresses most the points raised during the initial review process. However, it would be preferable if you included the methodological information referenced within this article rather than as a call back :

“for information on the genes and the promoter fragments assayed, see Experimental Procedures section in [ref]To avoid repetition and clearly inform the source of information, we have indicated the source in the text: 

“for information on the genes and the promoter fragments assayed, see Experimental Procedures section in [ref]

This would allow readers to access all the pertinent information without juggling between two references.

We have added S1 Table to indicate information on the genes and the promoter fragments assayed.

---

## [Editor Report · Decision Letter 2]

20 Nov 2019

Expressional artifact caused by a co-injection marker rol-6 in C. elegans

PONE-D-19-28634R2

Dear Dr. Kim,

We are pleased to inform you that your manuscript has been judged scientifically suitable for publication and will be formally accepted for publication once it complies with all outstanding technical requirements.

With kind regards,

Denis Dupuy, Ph.D.

Academic Editor

PLOS ONE
---

## [Editor Report · Acceptance letter]

25 Nov 2019

PONE-D-19-28634R2 

Expressional artifact caused by a co-injection marker *rol-6* in *C. elegans*

Dear Dr. Kim:

I am pleased to inform you that your manuscript has been deemed suitable for publication in PLOS ONE. Congratulations! Your manuscript is now with our production department. 

With kind regards,

on behalf of

Dr. Denis Dupuy 

Academic Editor

PLOS ONE